# Distinguishing the Relative Contribution of Environmental Factors to Runoff Change in the Headwaters of the Yangtze River

**Mengjing Guo** [1] , **Jing Li** [1,*], **Yongsheng Wang** [2], **Peng Bai** [3] and **Jiawei Wang** [1]

1    State Key Laboratory of Eco-hydraulics in Northwest Arid Region, Xi'an University of Technology, Xi'an 710048, Shaanxi, China
2    Key Laboratory of Regional Sustainable Development Modeling, Institute of Geographic Sciences and Natural Resources Research, Chinese Academy of Sciences, Beijing 100101, China
3    Key Laboratory of Water Cycle and Related Land Surface Process, Institute of Geographic Sciences and Natural Resources Research, Chinese Academy of Sciences, Beijing 100101, China
*    Correspondence: guomengjing@xaut.edu.cn

**Abstract:** The change in river flows at the basin scale reflects the combined influences of changes in various environmental factors associated with climatic and underlying surface properties. Distinguishing the relative contribution of each of these factors to runoff change is critical for sustainable water resource management, but it is also challenging. The headstream region of the Yangtze River, known as "China's Water Tower", has undergone a significant runoff change over the past decades. However, the relative contribution of environmental factors to runoff change is still unclear. Here, we designed a series of detrending experiments based on a grid-based hydrological model to quantify the combined influences of multiple environmental factors on runoff change and the relative contribution of an individual factor to runoff change. The results indicate that changes in climate and vegetation significantly increased water yield in the study basin over the past three decades, and the increase in water yield primarily came from the contribution from the upstream of the basin. On the basin scale, the change in precipitation dominated the runoff change that contributed up to 113.2% of the runoff change, followed by the wind speed change with a contribution rate of −15.1%. Other factors, including changes in temperature, relative humidity, sunshine duration (as a surrogate for net radiation), and albedo (as a surrogate for vegetation) had limited effects on runoff change, and the contribution rate of these factors to runoff change ranged from −5% to 5%. On spatial patterns, the influences of changes in some environmental factors on runoff changes were affected by elevation, particularly for temperature. The rising temperature had mixed effects on runoff change, which generally increased water yield at high altitudes of the basin but decreased water yield at low altitudes of the basin.

**Keywords:** climate change; river runoff; hydrological modeling; Yangtze River

## 1. Introduction

The science community now generally agrees that the Earth's climate is undergoing rapid changes in response to natural variability and increasing emission of greenhouse gases [1,2]. Climate change projection studies also indicated that climatic conditions in the future will significantly shift relative to the historical record in many regions of the world [3–5]. Climate change has profound impacts on global and regional water cycles by affecting the spatial-temporal patterns of climate variables such as precipitation, temperature, and radiation. Numerous studies reported that climate change is likely to influence the water cycle [6–9]. With global warming, more water will evaporate from

both land and oceans and be held in the air, increasing the amount of moisture circulating throughout the atmosphere [9]. Consequently, the Earth's climate system will become more unstable. Heavy rain events, heat wave events, and extreme droughts are likely to become more frequent in an intensifying water cycle, thereby triggering more natural disasters like flash floods, mudslides, soil erosion, landslides, and agricultural droughts [10–13]. In addition to climate change, land use/cover change (LUCC) has also altered the terrestrial water cycle in recent decades [14]. In many arid and semi-arid catchments, LUCC contributes more to runoff change than climate change [15,16]. Under the combined effects of climate change and LUCC, the global water cycle has been dramatically altered in the past decades, causing profound impacts on the global ecosystem and human society.

Many studies have reported the responses of terrestrial water balance variables to environmental change at both regional and global scales, such as changes in precipitation [17,18], evapotranspiration [19,20], soil moisture [21,22], and river discharge [23–25]. Among these water balance variables, change in river discharge has received much attention due to its importance to human societies, as well as economic and ecological systems. In many regions of the world, river runoff is the major source of water supply for humans. Therefore, change in river runoff is closely related to the survival and development of people who rely on rivers for water supply. River discharge, as the ultimate integrator of watershed hydrology, is influenced by both climate variability and land cover and land use change. Precipitation is generally the primary source of river discharge and has a crucial influence on variability in river discharge. Other climate variables besides precipitation, such as net radiation, temperature, relative humidity, and wind speed, can also affect river discharge indirectly by altering evapotranspiration. River discharge is also influenced by land use and land cover change, particularly in basins where intensive human activities exist. The influences of climate and landscape factors on river runoff are not independent but interact with each other. For example, precipitation and temperature directly determine the types and spatial distribution of land surface vegetation, and changes in vegetation on a large-scale will, in turn, affect the climate system by altering the energy budget of the land surface. Thus, it is complicated to quantify the relative importance of different climate and landscape factors on runoff change at the basin scale [26,27]. Furthermore, the relative contribution of the factors affecting runoff varies with climate, vegetation and soil conditions. Understanding of magnitudes, mechanisms, and interactions that control historical runoff changes is a prerequisite for predicting future runoff change, yet it is also complicated due to complex interactions and feedbacks between the factors influencing runoff.

Several approaches have been used for evaluating environmental change impacts on river runoff, among which the Budyko-based statistical model and the hydrological model are two commonly used approaches. The Budyko-based statistical model describes the first-order effect of changes in precipitation (P) and potential evapotranspiration (PET) on runoff [28]. This method assumes that the long-term water balance for a given basin depends primarily on P and PET and relates basin-mean runoff (evaporation) to P and PET with an empirical curve. However, this method is only suitable for large basins and for multiyear time scales, and it cannot provide the spatial distribution of runoff or ET simulations for a catchment [29]. Hydrologic models are simplified mathematical representations of the complex, dynamic, and non-linear processes of the rainfall-runoff transformation and are essential tools to assess runoff response to environmental change [30]. Hydrological models can be classified into lumped models and distributed models, depending on whether or not the model considers the spatial variability in the landscape characteristics and the forcing data [31]. They can also be classified into conceptual models and physically based models based on the extent of physical principles applied in the model [32]. For areas with rich ground-based observations, physically based models are preferred since they typically employ mathematic equations with clear physical meaning to describe the movement, storage, and transformation of water between the soil-vegetation-atmosphere continuum. However, these models often involve a variety of forcing data and a large number of parameters to be calibrated, which greatly hinders the application of such models, particularly in

gauge-sparse regions. In comparison, conceptual models have been the preferred tools in gauge-sparse river basins due to their low requirements of model inputs and easy parameterization [31].

A general approach for the assessment of environmental change impact on runoff is to divide the runoff time series into two periods: the "baseline or natural period" and the "impacted period", based on the identified abrupt change point(s) by a variety of statistical methods, such as non-parametric tests and double-mass curves [2]. In the "baseline period", change in runoff is mainly attributed to climate change, and the model parameters are calibrated in this period. In the "impacted period", however, change in runoff is considered to contain impacts arising from both climate and human-related activities. Using the parameters obtained from the "baseline period" and forcing data in the "impacted period", a runoff time series that only reflects the influences of climate change was modeled. The difference between the modeled and observed runoff in the "impacted period", which is attributed to other factors, such as human activities [27]. The above assessment method has two disadvantages. First, the abrupt change of runoff time series strongly depends on the used statistical methods, and different statistical methods may obtain different abrupt change points. In many cases, no abrupt change occurs in the runoff time series. Also, this assessment method cannot distinguish the influence of the change in an individual factor (e.g., precipitation or temperature) on runoff change [33].

The source area of the Yangtze River (above the Zhimenda station, see Figure 1) comprises 7.6% of the total area of the Yangtze River basin but contributes nearly 20% of the water volume of the Yangtze River [34,35]. Runoff regimes in this region have experienced a significant change over the past three decades (see Figure 2). In this study, we designed a series of detrending experiments based on a grid-based hydrological model to quantify the influences of climate and vegetation changes on runoff change in this region. We also discussed the advantage of the used attribution method and the sources of uncertainty in the assessment results. The findings of this study help us to develop the appropriate adaptive strategies to deal with the challenges of environmental change.

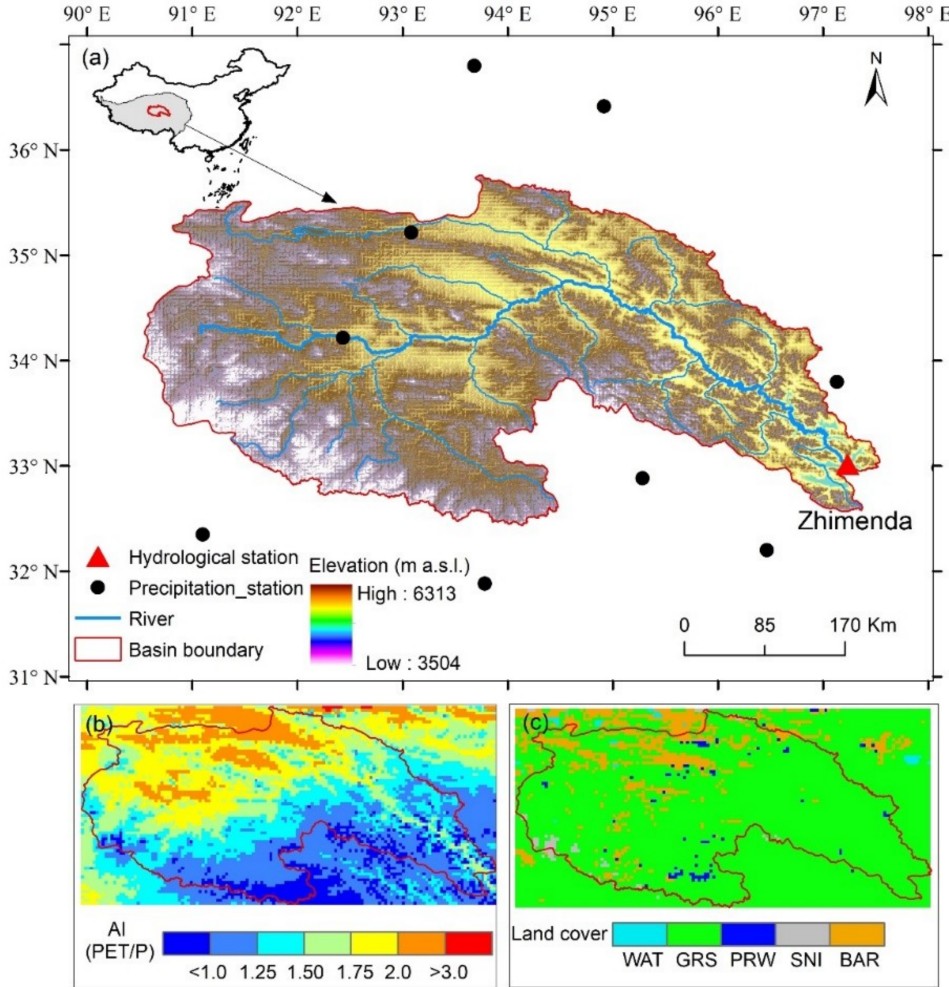

**Figure 1.** The location of the headwaters of the Yangtze River in the Tibet Plateau (grey polygon) and China (**a**) and the spatial patterns of aridity index (AI, the ratio of PET to P) and land cover types in the river basin (**b**,**c**). The abbreviations in Figure 1c denote different land cover types: WAT-water bodies, GRS-grasslands, PRW-permanent wetlands, SNI-snow and ice, and BAR-barren.

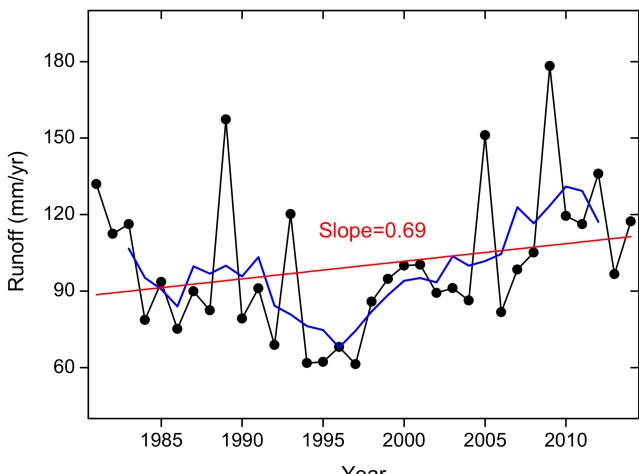

**Figure 2.** Interannual variation of observed runoff across the study basin. The red line indicates the fitted linear trend in the annual runoff time series, and the blue line shows the five-year moving average of the annual runoff time series.

## 2. Materials and Methodology

### 2.1. Study Area and Change in Runoff

The study area is located in headwaters of the Yangtze River in the central Tibetan Plateau (Figure 1), which covers an area of 137,704 km². The basin has a limited impact of human activities (e.g., large dams and water diversion projects) on river discharge since most of the basin is located in the "Three-River Headwaters" natural reserve, and the primary geomorphic type is high altitude mountains. The basin has large spatial heterogeneities in the terrain and climate. The elevations of the basin range from 3504 to 6313 m above the sea level (m a.s.l.), and the climate of the basin varies from humid to arid, with the aridity index (PET/P) ranging from 0.87 to 3.54 (Figure 1b). According to meteorological statistics from 1981 to 2014, the mean annual precipitation in the basin is 436 mm and decreases as the latitude increases, with 70–80% of precipitation occurring during the wet months (May–October). The average mean temperature of the basin is −4.4 °C, and the mean annual aridity index (PET/P) is 1.50. The land cover type is predominantly grasslands, covering 87% of the total basin area (Figure 1c), and the area of the glacier only accounts for 0.95% of the entire basin area.

The distribution of the meteorological stations in the basin is sparse and uneven, and there are only nine stations located in and around the basin (Figure 1a). We collected the daily meteorological data from the Chinese Meteorological Administration (http://data.cma.cn/), and the collected data include precipitation, temperature, relative humidity, wind speed, and sunshine hours. We interpolated gauge-based meteorological observations into grid-based datasets with a spatial resolution of 0.05° × 0.05° using a professional meteorological interpolation software, i.e., the Anusplin [36]. This software accounts for the influence of terrain change on meteorological variables [37,38]. We calculated the PET using the FAO-56 Penman–Monteith method [39] but replaced the default albedo parameter (0.23 for all land cover types) with the remotely sensed albedo data to reflect the influence of vegetation change on runoff. Here, the GLASS (Global Land Surface Satellite) albedo product was employed.

The observed annual runoff time series in the study basin shows opposite trends between the period 1981–1997 and the period 1998–2014, showing a significant decreasing trend in the first period and a significant increasing trend in the second period (Figure 2). Overall, the trend in annual runoff increased over the period 1981–2014, with a linear trend of 0.69 mm/yr. The mean annual runoff of the basin is 99.9 mm/yr. The impact assessment was not extended to the period before 1981 since the remotely sensed vegetation information has only been available since 1981.

### 2.2. Hydrological Model and Parameter Calibration

Here, we employed a widely used monthly water balance model (i.e., the abcd model [40]) to simulate the runoff response to environmental changes (Figure 3). The model has been applied in a large number of basins globally, with a variety of climate and landscape conditions [30,41–44]. The original abcd model contains four parameters (i.e., a, b, c, and d, see Table 1) and consists of four modules: ET, water storage, quick runoff, and base flow. Two cascaded storage layers are used in the model; the upper layer represents soil water storage, and the lower layer represents groundwater storage. We modified the model from a lumped version into a grid-based version to account for the spatial variability of forcing data. We also added a simple temperature-based snowfall-snowmelt module in the abcd model (hereafter called abcd-snow model) given the importance of snowfall and snowmelt processes in the study area.

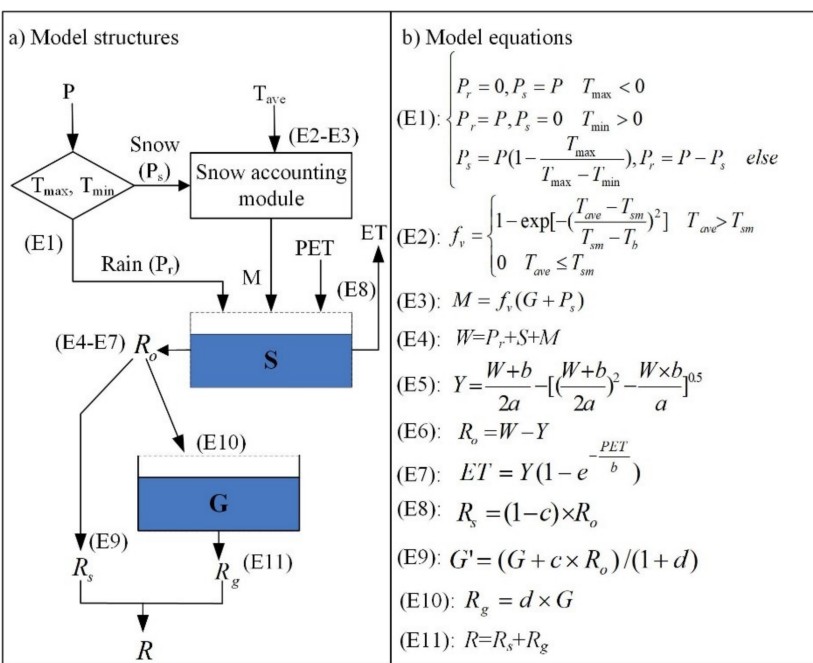

**Figure 3.** Model structures and key equations used in the abcd model that includes a snowfall-snowmelt module (hereafter called abcd-snow model). The descriptions of the variables and parameters are provided in Table 1.

**Table 1.** Parameters and variables used in the abcd-snow model.

| Parameter/Variable | Abb. | Meaning |
|---|---|---|
| Variable | $R_s$ | Surface runoff (mm) |
| | $R_g$ | Groundwater runoff (mm) |
| | $R_o$ | The water available for runoff (mm) |
| | $R$ | Total runoff that equals the sum of $R_s$ and $R_g$ (mm) |
| | $M$ | The snowmelt runoff (mm) |
| | $f_v$ | Snowmelt coefficient |
| | $W$ | Water availability that equals the sum of P and S (mm) |
| | $Y$ | The sum of ET and S (mm) |
| | $G$ | The routing storage (mm) |
| Parameter | $a$ | The propensity of runoff occur before the soils is fully saturated |
| | $b$ | The upper soil water storage capacity (mm) |
| | $c$ | Groundwater recharge coefficient |
| | $d$ | Groundwater runoff recession coefficient |
| | $T_{sm}$ | The critical temperature of snowmelt occurrence (°C) |
| | $T_b$ | The parameter that controls the velocity of snowmelt occurrence |

The modified model contained six parameters (Table 1) which were calibrated using the genetic algorithm by maximizing the goodness-of-fit between the observed runoff and the simulated runoff. The Kling–Gupta Efficiency (KGE) [45] was utilized as the objective function (see Section 2.4 for details). For each catchment, the available monthly runoff data (from January 1981 to December 2014) were split into two parts: the data from 1982–1997 were used for model calibration, and the data from 1998–2014 were used for model validation. The runoff data for 1981 were used for a warm-up of the model to reduce the impact of the initial values on the model calibration. The model forcing data included monthly P, PET, average temperature ($T_{ave}$), maximum temperature ($T_{max}$), and minimum temperature ($T_{min}$). Figure 3 provides the model structures and key equations used in the abcd-snow model. Descriptions of the free parameters and variables in the abcd model are listed in Table 1.

### 2.3. The Quantitative Method for the Contribution of Different Factors to Runoff Change

In the study basin, the influence of human activities on river discharge was limited. Therefore, change in runoff was primarily attributed to the impacts of changes in climatic and vegetation properties. In terms of the hydrological modeling framework used here, the climate variables that affected runoff change included precipitation, the variables used for the snow accounting module ($T_{max}$ and $T_{min}$), and the variables used for PET calculation, including $T_{ave}$, wind speed, relative humidity, and sunshine duration [39]. The remotely sensed albedo data were used to reflect the influence of vegetation change on runoff, which is one of the inputs of PET and directly impacts the surface net radiation estimates [39]. Here, we used the runoff simulations from the observed forcing variables as the benchmark (baseline scenario) and then applied a detrending method to identify the relative contribution of these forcing variables to runoff change. We can evaluate the combined influences of the environmental changes on runoff by detrending all the forcing variables and then running the model using the detrended forcing variables (see Section 3.2). We can also assess the contribution of a single forcing variable to runoff change using an individual detrended variable together with the other original variables as inputs (see Section 3.3). Take $T_{ave}$ as an example (see Figure 4), we first remove the annual trend of $T_{ave}$ using the detrending method. Second, we ran the hydrological model using the same forcing variables and parameters as the baseline scenario but replaced the observed $T_{ave}$ with the detrended $T_{ave}$. The difference in runoff simulations using original and detrended $T_{ave}$ was attributed to the influence of $T_{ave}$ change. Similarly, we can use similar steps to quantify the effects of variables other than $T_{ave}$ on runoff change individually. The contribution rate for each forcing variable of runoff can be computed as:

$$\begin{cases} CR(i) = 100 \times f_{adjust} \frac{\Delta R_i}{\Delta R}, \\ f_{adjust} = \frac{1}{\sum\limits_{i=1}^{i=n} \frac{\Delta R_i}{\Delta R}} \end{cases} \tag{1}$$

where $CR(i)$ denotes the contribution rate of the $i$-th forcing variable to runoff change (%), $\Delta R_i$ denotes the difference in mean annual runoff simulations obtained from the original and the detrended $i$-th models forcing (mm), $\Delta R$ denotes the difference in runoff simulations obtained from original and detrended forcing variables, $f_{adjust}$ is a scaling factor that ensures that the sum of CR is equal to 100%, and $n$ is the number of controlling factors of runoff ($n = 8$). A negative value of $CR(i)$ indicates that the change in the $i$-th forcing variable decreases the mean annual runoff compared to the baseline scenario, and vice versa. It should be noted that the value of $CR(i)$ can be larger or less than ±100% given that some forcing variables may have contrary effects on runoff change.

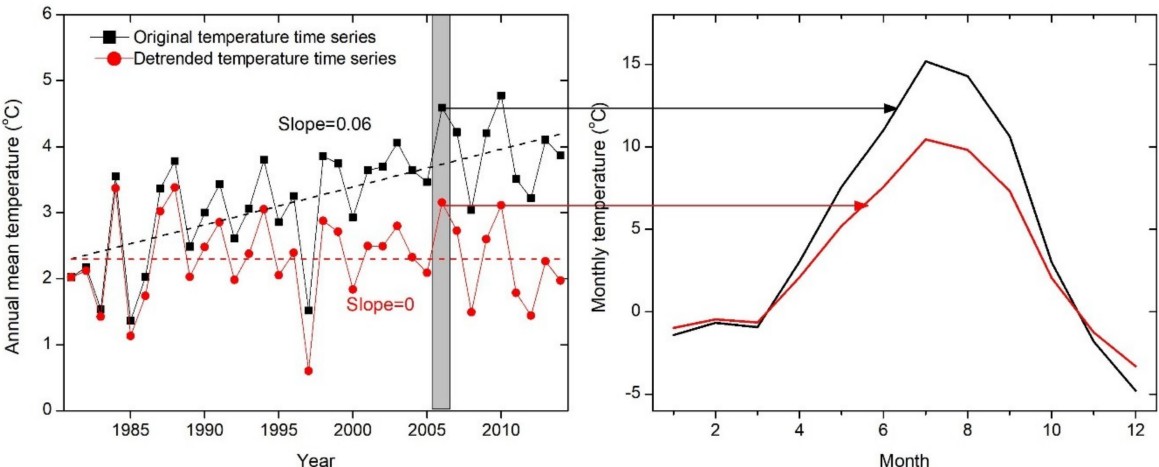

**Figure 4.** Schematic of the detrending method for a time series of annual average temperature. This method would eliminate the annual trend in observed climatic variables but would not impact seasonal variations in climatic variables.

*2.4. Model Performance Assessment*

We utilized three criteria to evaluate the agreement between the modeled and observed runoff time series, which are the relative bias (Bias), the coefficient of determination ($R^2$), and the Kling-Gupta Efficiency (KGE) [45]. The $R^2$ represents how much variation in the observations can be explained by the simulations [46–48]. The Bias indicates the average tendency of the simulated values to be larger or smaller than the observations [49], with the Bias = 0 being optimal. The KGE is typically used as a comprehensive assessment criterion to measure the agreement between the observed and the simulated values [50,51], which integrates three independent statistical criteria (i.e., linear correlation ($\gamma$), relative variability ($\alpha$) and bias ratio ($\beta$)) into a single multi-objective criterion [52,53]:

$$\text{KGE} = 1 - \sqrt{(1-\gamma)^2 + (1-\alpha)^2 + (1-\beta)^2} \text{ with } \alpha = \sigma_s/\sigma_o, \text{ and } \beta = \mu_s/\mu_o \qquad (2)$$

where $\mu_s$ and $\mu_o$ are the averages for the simulations and the observations, respectively, and $\sigma_s$ and $\sigma_o$ are the standard deviations for the simulations and the observations, respectively. The KGE ranges from negative infinity to 1, and an optimal KGE is equal to 1.

## 3. Results

*3.1. Changes in Climatic and Vegetation Properties*

These model forcings had different influence mechanisms on runoff from the perspective of the hydrological cycle. Precipitation directly impacted runoff generation, while the other forcing variables regulated runoff change primarily by affecting evapotranspiration and/or soil moisture. Figure 5 shows the interannual variability and trends in six climatic variables and two vegetation indexes. We identified the significance of the trend in the time series with the non-parametric Man-Kendall (MK) statistical test [54,55]. The results indicate that the climate of the study basin underwent a warming and wetting trend (Figure 5a). There was a significant increasing trend ($p < 0.05$ and slope > 0) in annual precipitation and average temperature during the period 1981–2014, and the trends in the two variables were 2.32 mm/yr and 0.07 °C/yr, respectively. The rate of warming in the study basin was more than twice the national average rate (~0.03 °C/yr) [56]. The results are in line with some previous studies [57–59]. Unlike the trends in precipitation and temperature, the relative humidity and wind speed show an insignificant decreasing trend ($p > 0.05$ and slope < 0), and the trends in the two climatic variables were −0.08%/yr and −0.013 m/s/yr, respectively. Moreover, the remotely sensed NDVI (Normalized Difference Vegetation Index) and albedo exhibit an insignificant increase over the period 1981–2014. The calculated PET and aridity index (PET/P) using these variables presented opposite trends: the trend in the PET time series significantly increased, while the trend of the aridity index significantly decreased (Figure 5a).

On the spatial pattern of trends in these variables (Figure 6), precipitation shows a larger trend in the upstream of the basin than that in the downstream of the basin (Figure 6a). Trends in temperature and wind speed were related to elevation change, and the high-altitude regions tended to exhibit larger trends (Figure 6b,e). Trends in other variables had no apparent spatial characteristics.

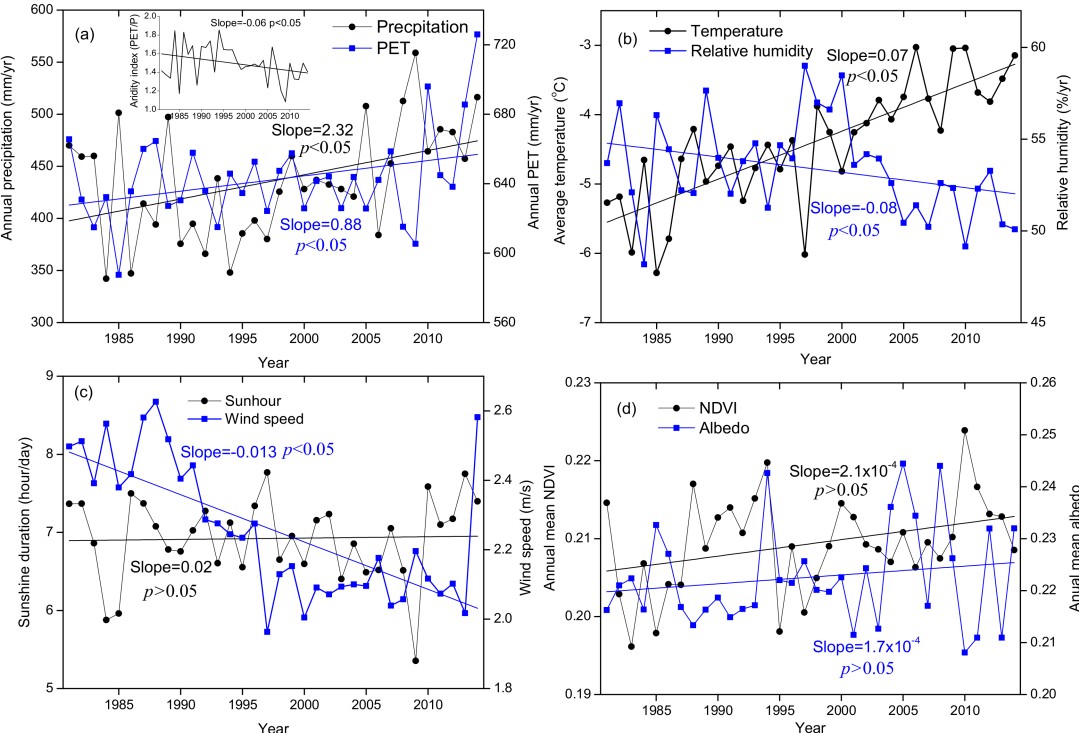

**Figure 5.** Interannual variability and trends in climatic and vegetation properties across the study basin during the period 1981–2014. In each panel, the slope of the fitted line indicates the trend in time series, and the trend is statistically significant when the *p*-value is less than 0.05. The subfigure in Figure 4a presents the variation of annual aridity index (PET/P) during the period 1981–2014.

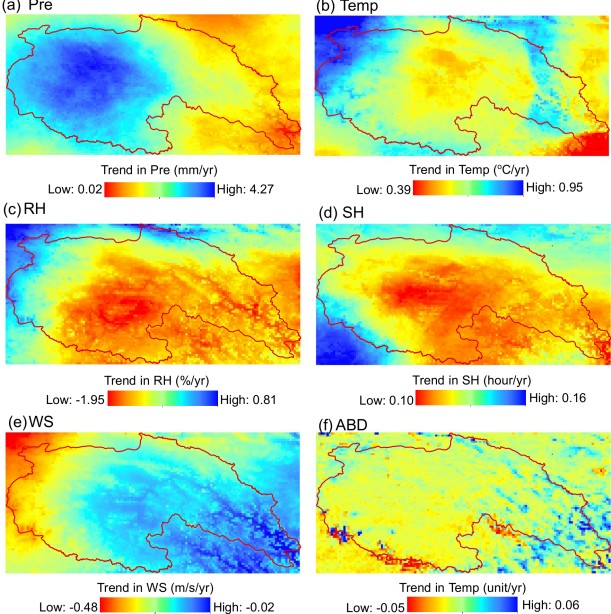

**Figure 6.** Spatial pattern of trends in climatic and vegetation characteristics in the study basin over the period 1981–2014. The abbreviations in the figure denote different environmental variables: Pre-precipitation, Temp-mean temperature, RH-relative humidity, SH-sunshine hour, WS-wind speed at 2 m above the ground, and ABD-albedo.

### 3.2. Combined Influences of Environmental Changes on Runoff

It is necessary to evaluate the model's runoff simulation ability before applying the hydrological model to assess the impact of environmental changes on runoff. Figure 7 presents the model performance in the runoff simulations. In general, the model reproduced the monthly runoff time series well both in the calibration period (1982–1997) and in the validation period (1998–2014). The three assessment criteria, i.e., $R^2$, Bias, and KGE, were 0.85, 1.4%, and 0.92, respectively, during the calibration period; these criteria were 0.87, 3.2%, and 0.91, respectively, during the validation period. Therefore, the abcd-snow model can be used as an effective tool to assess the influences of environmental changes on runoff. Furthermore, the spatial pattern of runoff simulations was similar to that of the aridity index (Figure 8a), showing a higher value in the northwest of the study area and a lower value in the southeast of the study area.

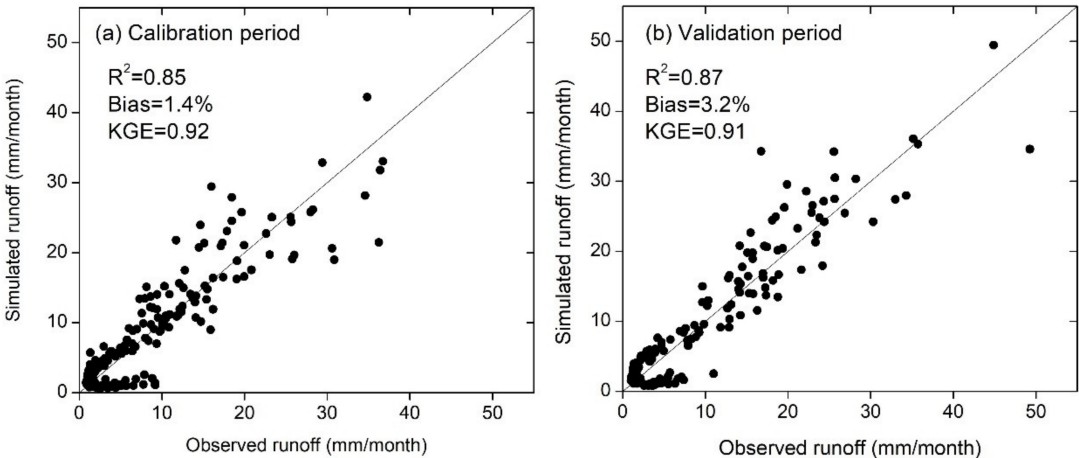

**Figure 7.** Comparison of the observed and simulated runoff during the calibration period (1982–1997) (**a**) and the validation period (1998–2014) (**b**).

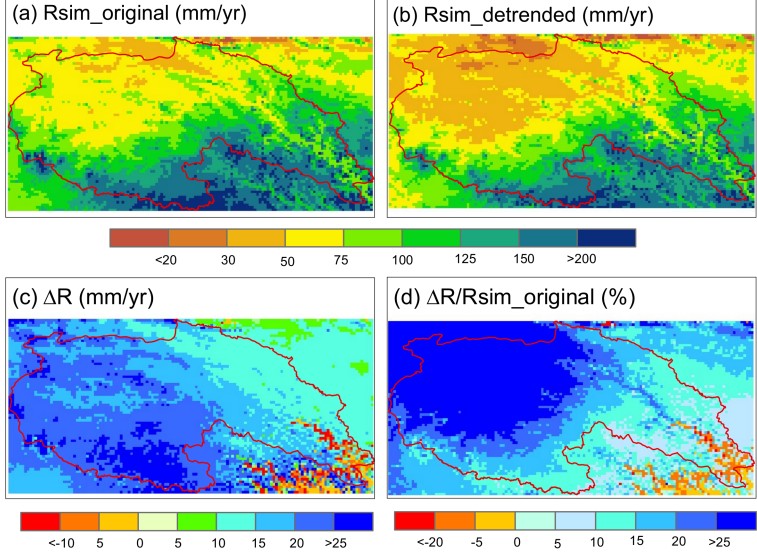

**Figure 8.** Comparison of simulated runoff across the study basin using the original (Rsim_original) and detrended (Rsim_detrended) forcing variables (**a**,**b**). Here, the detrending was performed for all the forcing variables used in the abcd-snow model. (**c**) the difference in the mean annual runoff simulations obtained from the original and detrended forcing variables (ΔR). (**d**) ΔR as the percentage of Rsim_original across the study basin (ΔR/Rsim_original).

Here, we performed the detrending for all forcing variables and ran the model using these detrended forcing variables. The difference in runoff simulations ($\Delta R$) obtained from the original and detrended forcing variables can be attributed to the combined influences of the environmental changes on runoff. As shown in Figure 8c, the influences of environmental change on runoff yield are significantly different on the spatial pattern. Larger $\Delta R$ values (>25 mm/yr) were located south of the basin, and lower values (<−10 mm/yr) appeared in the southeast of the basin. $\Delta R$ as the percentage of the observed mean annual runoff (Robs) exceeded 25% in the northwest of the basin and was less than −20% in the southeast of the basin (Figure 8d). The basin-average $\Delta R$ was 18.8 mm/yr, accounting for 18.8% of the mean annual runoff of the basin (99.9 mm/yr). This indicates that environmental changes significantly increased water yield in the study basin over the period 1981–2014, compared to the "zero-trend" scenario of environmental changes, and the increase in runoff primarily comes from the contribution from the upstream (the lower left part) of the basin (Figure 8c).

### 3.3. The Relative Contribution of a Single Forcing Variable on Runoff Change

Here, we evaluated the influence of a single forcing variable on runoff change. Section 2.3 provided a detailed description of the assessment steps. It should be noted that the detrending was performed for the three temperature variables (i.e., the maximum, minimum, and average temperatures) simultaneously, considering the strong correlation between them. The left column of Figure 9 shows the difference in the mean annual runoff obtained from the baseline scenario and the detrending scenario of a single forcing variable, and the contribution rate of a single forcing variable to runoff change is provided in the right column of Figure 9.

Among these forcing variables, change in precipitation was the largest contributor to runoff change, which contributed 113.2% of runoff change in the whole basin (Table 2). Also, the contribution rate of precipitation to runoff change demonstrates a decreasing trend along the flow direction of the river. Wind speed change was the second largest contributor to runoff change in terms of the absolute value of the contribution rate, which generally decreased runoff yields during the period 1981–2014 and contributed −15.1% of runoff change (Table 2). On the spatial pattern, runoff response to wind speed change gradually decreased from upstream to downstream of the basin (Figure 9j). The change in temperature had mixed impacts on runoff change, which generally increased water yield at higher altitudes of the basin but decreased water yield at lower altitudes of the basin (Figure 9c). Overall, the change in temperature increased runoff yields: the basin average contribution rate of temperature changed to runoff change was 3.8% for the whole basin. Similarly to the runoff response to temperature change, the albedo change also had mixed impacts on runoff change, with a basin-average contribution rate of 1.9%, and the degree of influence seems to be related to elevation. Changes in relative humidity and sunshine duration had a limited influence on runoff change, and they contributed 1.8% and 0.9% of runoff change, respectively, on the basin scale.

In summary, change in precipitation dominated runoff change in the study basin, which contributed 113.2% of runoff change on average. Similar conclusions were drawn in previous research [57,60–62]. Changes in temperature and albedo had mixed effects on runoff change, and their impacts on runoff change were related to elevation, and they overall contributed 3.8% and 1.9% of runoff change on the basin scale, respectively. Changes in wind speed, relative humidity, and sunshine duration overall decreased the water yield during the study period, and they contributed −15.7%, −2.5%, and −1.3% of runoff change on the basin scale, respectively.

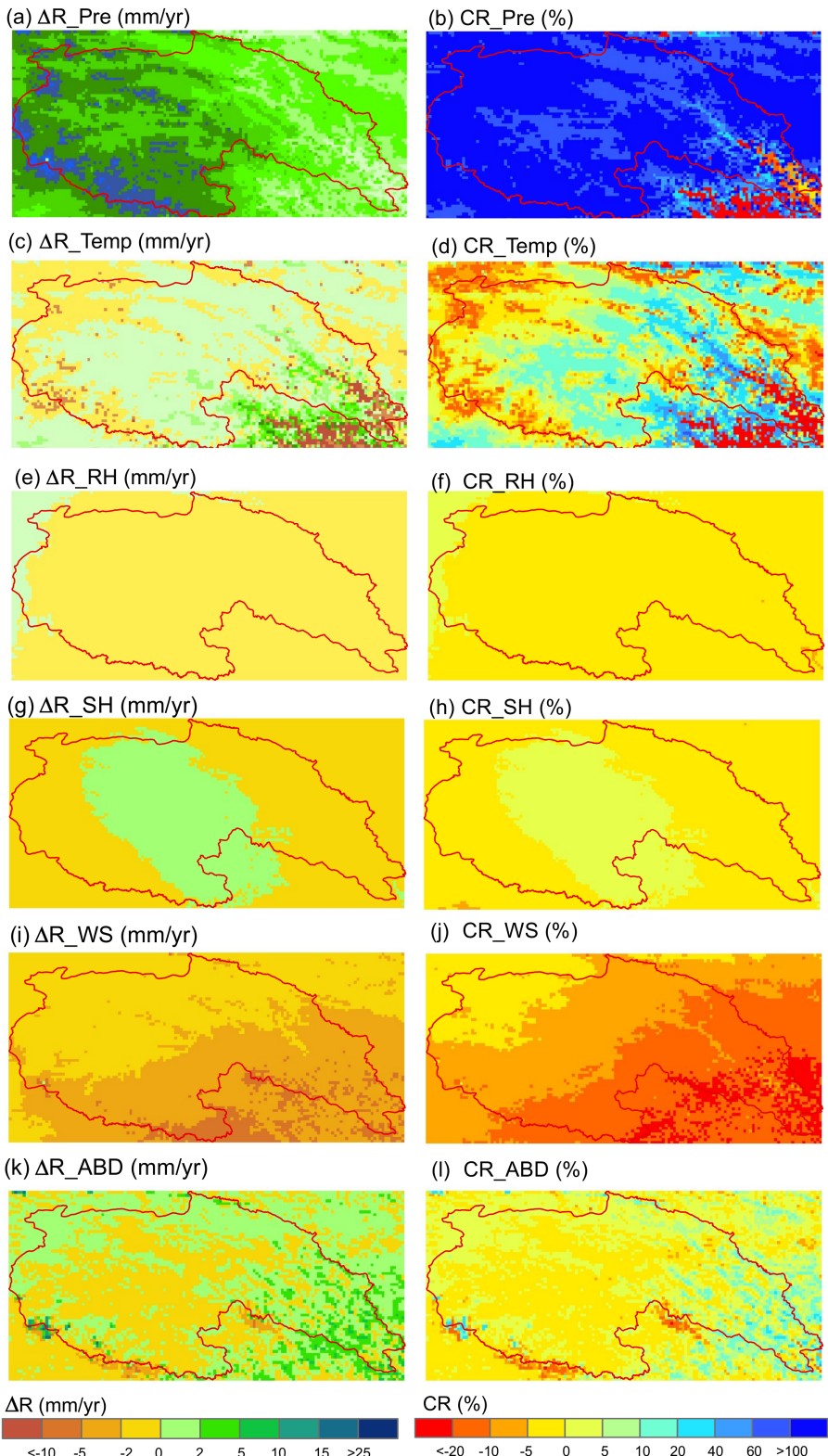

**Figure 9.** The difference in runoff simulations obtained from the original and detrended forcing data (left column), and the contribution of each forcing variable to runoff change (the right column). Unlike Figure 8c, the detrending was only performed for a specific forcing variable. The meaning of abbreviations after "ΔR_" or "CR_" are as follows: Pre—precipitation, Temp—temperature, RH—relative humidity, SH—sunshine hour, WS—wind speed, and ABD—albedo.

**Table 2.** The basin-average ΔR and CR under different detrending scenarios. ΔR denotes the difference in mean annual runoff simulations obtained from the baseline scenario and the detrending scenario for a single forcing variable (or all forcing variables) during the period 1981–2014. CR denotes the contribution rate of a single forcing variable to runoff change during the period 1981–2014.

| Detrending Scenario | Mean Annual Runoff (mm/yr) | ΔR (mm/yr) | CR (%) |
|---|---|---|---|
| Baseline scenario | 102.4 | - | - |
| All forcing variables | 83.6 | 18.8 | - |
| Precipitation only | 84.4 | 18.0 | 113.2 |
| Temperature only | 101.8 | 0.6 | 3.8 |
| Relative humidity only | 102.8 | −0.4 | −2.5 |
| Sunshine duration only | 102.6 | −0.2 | −1.3 |
| Wind speed only | 104.8 | −2.4 | −15.1 |
| Albedo only | 102.1 | 0.3 | 1.9 |

## 4. Discussions

Numerous studies focused on assessing the impacts of environmental changes on runoff [26,27,61,63]. Investigating the effects of environmental changes on runoff usually involves the application of hydrologic models [64,65]. The choice of a hydrological model may have an impact on the final assessment results. Here, a widely used monthly hydrological model (i.e., abcd model) was employed. We modified the model into a grid-based version and added a temperature-based snow accounting module in the model. The calibration and validation results indicated that the model can well reproduce the observed monthly runoff at the outlet of the basin (Figure 7). However, some potential factors affecting runoff were not included in the current hydrological model, such as glaciers and permafrost, which cover ~0.95% and ~75% of the total area of the basin, respectively. The glacial meltwater directly supplies river flows, and permafrost limits moisture exchanges among different soil layers and reduces soil infiltration capacity, thereby altering flow regimes [66–68]. The effect of permafrost degradation on water yield is complicated under warming [21,69,70]. On the one hand, warming could promote the release of moisture in frozen soils and potentially increase water yield. On the other hand, warming could reduce the thickness of frozen soils and enhance soil water storage capacity, thereby reducing water yield [34,71]. However, the area of glaciers accounts for a small proportion of the total area of the basin (~0.95%) [57], and permafrost degradation does not necessary increase water yield [21]. Thus, the meltwater from glaciers and permafrost accounts for a small proportion of the total runoff, especially for the large river basin [34,57,61]. From the perspective of model simplification, ignoring glacial and frozen soil modules in hydrological models does not have a significant influence on the assessment results. In addition, although we attributed runoff change in the study basin to the combined effects of the changes in climate and vegetation, it is highly possible that human activities did play a role in runoff change as well. The study area has no large dams or water transfer projects but holds rich alpine grasslands and has long been a Tibetan pastoral area [72]. Thus, human activities (e.g., agriculture irrigation and grazing) also have an impact on runoff. However, according to previous studies, the extent of human activity impacts on runoff is small relative to climate change in this area [26,57,62,69]. The degree to which human activities impact runoff change in this area will be the focus of a future study.

In this study, we designed a series of detrending experiments for forcing variables based on a hydrological model to quantify the relative contribution of different environmental factors to runoff change in the source regions of the Yangtze River. This method had two distinct advantages over traditional assessment methods. First, this method did not require dividing the time series into two different periods and avoided uncertainty in the abrupt change test of the runoff time series. This allowed this method to be applied to the basins where no abrupt change occurred in the runoff time series. Second, this method was suitable for assessing the combined effects of all the forcing

variables on runoff change and the contribution of an individual forcing to runoff change, which is difficult to achieve with traditional assessment methods.

## 5. Conclusions

Over the last three decades, observed runoff from headwaters of the Yangtze River has exhibited a significant increase. However, the relative contribution of different environmental factors to increases in runoff has been unclear to date. In this study, we quantified the contributions of multiple environmental factors to runoff change in the headwaters of the Yangtze River. The main findings of this study are as follows:

(1) The climate in the study area underwent significant changes over the period 1981–2014, characterized by a significant increase in precipitation and temperature, and a significant decrease in wind speed.

(2) Changes in climate and vegetation significantly increased water yield in the study basin over the past three decades, and the increased water yield was primarily due to the contribution from the upstream (the lower left part) of the basin.

(3) On the basin scale, precipitation change was the largest contributor to runoff change over the study period, followed by wind speed change, and they contributed 113.2% and −15.1% of runoff change, respectively. The contribution rates from other factors other than precipitation and wind to runoff change were limited and ranged from −5% to 5%. Changes in temperature and albedo had mixed effects on runoff change, and their influences on runoff were associated with elevation.

**Author Contributions:** J.L. performed the research design; M.G. analyzed the data and wrote the draft; J.W. collected the hydrological and meteorological data; Y.W. and P.B. edited the draft and provided constructive suggestions to improve the paper.

**Funding:** This research was supported by the National Key Research and Development Program of China (No. 2016YFC0401402) and the Natural Science Foundation of China (No. 41601034 and 41807156).

**Conflicts of Interest:** The authors declare no conflict of interest.

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
