# Peer review of "Distinguishing the Relative Contribution of Environmental Factors to Runoff Change in the Headwaters of the Yangtze River"

_water, doi:10.3390/w11071432_

Round 1

Reviewer 1 Report

The article Distinguishing the relative contribution of environmental factors to runoff change in the headwaters of the Yangtze River (ID: water-545770) is presenting the contribution of different environmental factors to runoff change in an area that is very important especially worldwide and then to China (one of the largest rivers: the Yangtze River). The article is well-written and argumented accordingly. The English of the manuscript is very good.

At this stage, I recommend a Minor Revision, as there are some minor things to be considered and are mentioned below:

L43: replace “to intensify” with “change/influence”

L47-48: I think here you can also add soil erosion and landslides (of course, adding some references)

L56: those references should be in [   ]

Also, within the Introduction section you should also refer to the hydrologic response units (HRU).

L117: correct will be “we will also discuss” (as you are planning to do it)

Fig. 1. I think that instead of “±” you meant to insert the North Arrow? Please, try to use earth tones for the DEM, as the low altitudes represented in white, it looks like you have some gaps in the DEM, which you do not. And using the colour ramp that you used, it makes it hard to locate the hydrological station (took me about 2-3 minutes). Another thing that you should correct is the caption 1a. the location of the study area where? In China ? or within the entire Yangtze River basin? Please, specify. You should also include a small caption with the whole China. And since you have located on the map the precipitation stations, I think you should also include their names within the map.

Figures 5 and 6 need to be placed a bit earlier in the text; same for Figure 9.

I think that the caption of Figure 8 needs to be a bit smaller.

Author’s contribution section is missing. Please, correct.

References: why is the reference number doubled? Please, correct.

Kind regards,

Good luck with the review.

Author Response

The comments by reviewer #1

The article Distinguishing the relative contribution of environmental factors to runoff change in the headwaters of the Yangtze River (ID: water-545770) is presenting the contribution of different environmental factors to runoff change in an area that is very important especially worldwide and then to China (one of the largest rivers: the Yangtze River). The article is well-written and argumented accordingly. The English of the manuscript is very good.  

At this stage, I recommend a Minor Revision, as there are some minor things to be considered and are mentioned below:

 Answer: thanks very much for your work on our manuscript. The suggestions and comments are really helpful for improving our manuscript. We have accepted practically all the comments and suggestions and implemented them in the revised manuscript as shown below. The 

L43: replace “to intensify” with “change/influence”

Answer: done!

L47-48: I think here you can also add soil erosion and landslides (of course, adding some references)

Answer: done!

L56: those references should be in [   ]. Also, within the Introduction section you should also refer to the hydrologic response units (HRU).

Answer: done!

L117: correct will be “we will also discuss” (as you are planning to do it)

Answer: done!

Fig. 1. I think that instead of “±” you meant to insert the North Arrow? Please, try to use earth tones for the DEM, as the low altitudes represented in white, it looks like you have some gaps in the DEM, which you do not. And using the colour ramp that you used, it makes it hard to locate the hydrological station (took me about 2-3 minutes). Another thing that you should correct is the caption 1a. the location of the study area where? In China ? or within the entire Yangtze River basin? Please, specify. You should also include a small caption with the whole China. And since you have located on the map the precipitation stations, I think you should also include their names within the map.

Answer: the suggestion is taken. The Figure 1 was distortion when the manuscript in the word version was transferred to PDF version, resulting in a series of strange symbols like “±” and gaps in the DEM. In the revised M/S, we have mapped the Figure 1 as your suggestions: 1) using the international cartographic standards to recolor the elevations; 2) enlarging the symbol of hydrological station; 3) including an additional map showing the location of the study area in the Tibet Plateau and China. Also, we didn’t add the names of the precipitation stations since some sites have long Chinese name symbols, and adding them would cover other parts of the map.

Figures 5 and 6 need to be placed a bit earlier in the text; same for Figure 9.

Answer: done!

I think that the caption of Figure 8 needs to be a bit smaller.

Answer: done!

Author’s contribution section is missing. Please, correct.

Answer: done!

References: why is the reference number doubled? Please, correct.

Answer: done!

Reviewer 2 Report

The paper analyzes main environmental factors and their relative contributions to the observed changes of runoff in the upper reaches (headwaters) of the Yangtze River. In my opinion the submission is interesting and deserves attention, especially when considering the climate change scenarios projected for that area. However, there exist some shortcomings, which require improvements prior to the final acceptance of the submission for publication. They are as follows:

1. Figure 1 – location of the study area in China should be shown on an additional map.

2. Figure 1 – elevations should be expressed in meters above sea level (m a.s.l.). Please correct.

3. Figure 1 – colors showing elevations should be used in accordance to the international cartographic standards. As the elevations of the study area are higher than 3500 m a.s.l., green colors (used for plains) should be avoided. Instead, shades of brown, red and orange should be applied. Please correct.

4. Figure 1 – what are the white areas shown on the map? An explanation of that signature should be added in the legend.

5. Figure 1 – the signature “!” used to mark precipitation stations on the map looks quite strange, so it is suggested to change it and use a dot or triangle symbol instead. Similarly, the signature “#” used for marking hydrological station Zhimenda should be replaced by another symbol.

6. Figure 1 – what is the meaning of the signature “” drawn in the upper right corner of the map? Is it the North arrow? If yes, it should be replaced by the more appropriately looking North arrow symbol.

Generally, it is recommended to accept the submission for publication after the above-mentioned amendments are made.

Author Response

Responses to Reviewer#2’s comments:

The paper analyzes main environmental factors and their relative contributions to the observed changes of runoff in the upper reaches (headwaters) of the Yangtze River. In my opinion the submission is interesting and deserves attention, especially when considering the climate change scenarios projected for that area. However, there exist some shortcomings, which require improvements prior to the final acceptance of the submission for publication. They are as follows:

 Answer: thanks very much for your work on our manuscript. The suggestions and comments are really helpful for improving our manuscript. We have accepted practically all the comments and suggestions and implemented them in the revised manuscript as shown below. The revised M/S is attached and the corresponding modifications in the test have been marked yellow.

1.     Figure 1 – location of the study area in China should be shown on an additional map.

 Answer: the suggestion is taken. We added an additional map to show the location of the study area in the Tibet Plateau and China.

2.     Figure 1 – elevations should be expressed in meters above sea level (m a.s.l.). Please correct.

 Answer: done!

3.     Figure 1 – colors showing elevations should be used in accordance to the international cartographic standards. As the elevations of the study area are higher than 3500 m a.s.l., green colors (used for plains) should be avoided. Instead, shades of brown, red and orange should be applied. Please correct.

Answer: the suggestion is taken. We used the international cartographic standards to recolor the elevations.

4.     Figure 1 – what are the white areas shown on the map? An explanation of that signature should be added in the legend.

Answer: The Figure 1 was distortion when the manuscript in the word version was transferred to PDF version, resulting in a series of strange symbols like “±” , “!” , and gaps in the DEM. We have corrected the figure in the revised M/S. 

     5.     Figure 1 – the signature “!” used to mark precipitation stations on the map looks quite strange, so it is suggested to change it and use a dot or triangle symbol instead. Similarly, the signature “#” used for marking hydrological station Zhimenda should be replaced by another symbol.

Answer: please see the response above.

    6. Figure 1 – what is the meaning of the signature “” drawn in the upper right corner of the map? Is it the North arrow? If yes, it should be replaced by the more appropriately looking North arrow symbol.

 Answer: please see the response above.

   Generally, it is recommended to accept the submission for publication after the above-mentioned amendments are made.

Answer: Thanks again for your work.
